# Graph-Guided Reconstruction Diffusion for Multivariate Time Series Anomaly Detection

## Abstract

Time series anomaly detection often faces challenges such as non-stationarity and trends. Recently, unsupervised learning methods combined with generative models have shown promising prospects in this field, especially the application of multi-resolution technology in anomaly detection has achieved certain results. However, existing models usually ignore the correlations among different features in time series data and the rich multi-resolutional knowledge contained in the original data. To solve this problem, this paper proposes a new Model, **G**raph **G**uided **R**econstruction **D**iffusion Model (GGRD). GGRD is an end-to-end unsupervised anomaly detection model based on reconstruction. It adopts overlapping sliding Windows to sample multi-resolution data and integrates the similarity prior in the data into the **G**raph-**G**uided **A**ttention (GGA) mechanism, thereby effectively dealing with complex characteristics such as non-stationarity and cross-variable correlations of time series. The experimental results show that GGRD significantly outperforms the existing methods on multiple real datasets. Code is available at https://anonymous.4open.science/r/GGRD-806F/.

## 1 Introduction

Time series anomaly detection is a fundamental problem with applications in industrial monitoring, healthcare, and cybersecurity. Unlike other modalities, time series often exhibit trends, seasonality, and—most critically—non-stationarity, which obscure the boundary between normal and abnormal patterns (Wen et al., 2022). In multivariate settings, the challenge is amplified by evolving cross-feature dependencies, where correlations emerge, vanish, or shift over time (Wang et al., 2023). These properties render conventional anomaly detection methods insufficient for real-world time series.

Recently, diffusion-based generative models (Chen et al., 2023; Wang et al., 2023) have shown promise for anomaly detection. By corrupting time series with noise and reconstructing the original signal, they provide an unsupervised framework where reconstruction error serves as the anomaly score. Yet, existing approaches face three fundamental limitations. (i) **Non-stationarity across resolutions.** MODEM (Zhong et al., 2025) incorporates multi-resolution decomposition, but its reliance on non-overlapping pooling introduces staircase artifacts and discards fine-grained information. Other methods such as MG-TSD (Fan et al., 2024) and MR-Diff (Shen et al., 2024) attempt multi-granularity supervision or progressive denoising, but still yield coarse, unstable representations. (ii) **Inefficient iterative denoising.** ImDiffusion (Chen et al., 2023) and DiffAD (Xiao et al., 2023) adopt multi-step denoising, which incurs high computational cost and error accumulation, limiting real-time applicability. (iii) **Missing explicit modeling of feature dependencies.** Anomaly Transformer (Xu et al., 2021) focuses on temporal association discrepancies but overlooks cross-feature dynamics. Graph-based models such as MTAD-GAT (Zhao et al., 2020) and GDN (Deng & Hooi, 2021) employ fixed or weakly adaptive graphs, while the Diffusion Graph Model (Lan et al., 2025) introduces anomaly-aware edges without capturing evolving correlations.

To address these gaps, we propose the *Graph-Guided Reconstruction Diffusion model (GGRD)*, an end-to-end framework for unsupervised anomaly detection. GGRD introduces three innovations: (i) a sliding-window averaging mechanism for smooth and stable multi-resolution decomposition, (ii) a one-step reconstruction strategy that replaces iterative denoising to improve efficiency and robustness, and (iii) a Graph-Guided Network (GGN) equipped with Graph-Guided Attention (GGA),

which injects similarity-based priors into the attention mechanism to capture dynamic cross-feature dependencies.

In summary, the contributions of this paper are:

- Proposes **GGRD**, a diffusion-based framework that integrates smooth multi-resolution decomposition with efficient one-step reconstruction.
- Introduces **GGN** with Graph-Guided Attention to explicitly model dynamic cross-feature dependencies.
- Demonstrates state-of-the-art performance and efficiency on multiple real-world benchmarks.

## 2 RELATED WORK

Research on time series data mainly focuses on classification, prediction, imputation, and anomaly detection(Jin et al., 2024). Among them, anomaly detection in time series has attracted considerable attention in both industry and academia(Zamanzadeh Darban et al., 2024), especially for multivariate time series data. The methods for multivariate time series anomaly detection can be roughly divided into three categories(Zhang et al., 2025): early traditional statistical methods, traditional machine learning methods, and deep learning methods.

Statistical methods mainly utilize statistical knowledge and some statistical indicators to achieve anomaly detection tasks, such as ARIMA(Yaacob et al., 2010) and COPOD(Li et al., 2020). Traditional machine learning methods such as PCA(Shyu et al., 2003), kNN(Ramaswamy et al., 2000), and IForest(Liu et al., 2008) use various linear transformations, proximity measures, or outlier detection algorithms to effectively identify abnormal behaviors. However, these methods cannot effectively capture the complex patterns of time series data.

In recent years, with the rapid development of deep learning, a large number of deep learning-based time series anomaly detection models have been proposed and achieved remarkable results in different scenarios, such as VAE(Kingma & Welling, 2013), LSTM-AD(Malhotra et al., 2015), GDN(Deng & Hooi, 2021), Anomaly Transformer(Xu et al., 2021), etc. It is worth noting that Diffusion Models have demonstrated outstanding performance in image generation(Xu & Chi, 2024; Luo et al., 2024; Epstein et al., 2023), prompting researchers to introduce them into time series anomaly detection tasks. ImDiffusion(Chen et al., 2023) uses diffusion models to mask and fill time series and combines ensemble strategies to enhance the robustness of anomaly detection, being one of the earliest works to apply diffusion models to time series anomaly detection. DiffAD(Xiao et al., 2023) proposes a new denoising diffusion-based imputation method and uses a density ratio-based strategy to flexibly select normal observations, thereby reducing the interference of dense anomaly regions on the model. D3R(Wang et al., 2023) proposes a dynamic decomposition and diffusion reconstruction framework for non-stationary time series, which significantly reduces the impact of drift on detection accuracy by achieving dynamic decomposition of stable and trend components and using noise diffusion to control the information bottleneck externally.

To fully utilize the information in the data, some multi-resolution methods have been applied to time series tasks. For example, MG-TSD(Fan et al., 2024) uses multi-granularity guided loss to enhance prediction performance to address the instability challenge caused by randomness. MR-diff(Shen et al., 2024) uses seasonal trend decomposition and a coarse-to-fine non-autoregressive method to solve prediction tasks. MODEM(Zhong et al., 2025) designs a multi-resolution decomposable diffusion model for the anomaly detection task of non-stationary time series, with the core being a coarse-to-fine diffusion process and a frequency domain enhanced decomposition network, which can capture long-term trends and short-term fluctuations at different time scales, thereby effectively distinguishing anomalies from non-stationary patterns.

The above-mentioned methods have promoted the development of the field of anomaly detection from the perspectives of statistics, generative models, and multi-resolution modeling. In this work, we for the first time explicitly introduce the correlation of different features of time series in different temporal granularity into the modeling. We consider multi-resolution data modeling and propose a novel reconstruction diffusion model to achieve the modeling ability of complex time series relationships.

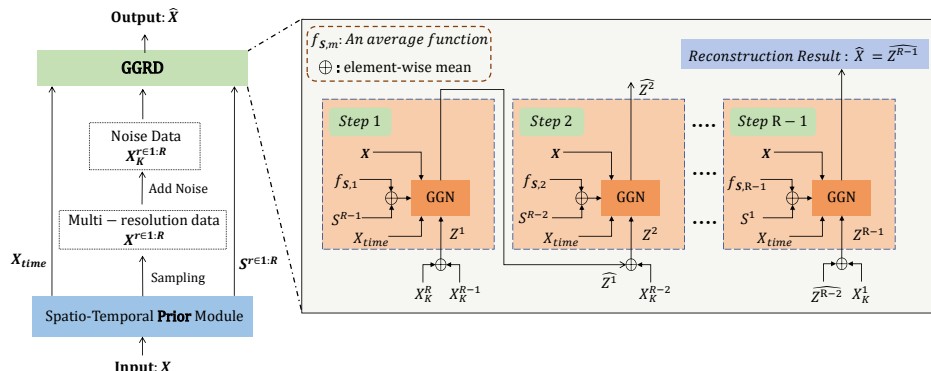

Figure 1: The overall structure of our model is summarized, mainly including STPM and GGRD.

## 3 PRELIMINARY

The goal of multivariate time series anomaly detection is to identify time steps at which the observed values deviate from normal behavior. Formally, let $\mathbf{X} \in \mathbb{R}^{T \times D}$ denote the input time series, where $T$ is the number of time steps and $D$ is the number of features. The anomaly labels are represented as $\mathbf{Y} \in \{0, 1\}^T$, where $y_t = 1$ indicates that the observation at time step $t$ is anomalous and $y_t = 0$ otherwise. In the unsupervised setting considered here, labels $\mathbf{Y}$ are unavailable during training and are used only for evaluation.

A common approach to unsupervised anomaly detection is to learn a generative model that captures the distribution of normal time series. The pipeline typically consists of three stages: (i) **Training:** fit a generative model $p_\theta(\mathbf{X})$ (or its conditional variant) using unlabeled historical data assumed to be mostly normal; (ii) **Reconstruction or prediction:** given a test input $\mathbf{X}$, obtain a reconstructed (or predicted) version $\hat{\mathbf{X}}$ using the generative model; (iii) **Scoring and thresholding:** compute an anomaly score $s_t$ for each time step, e.g., via reconstruction error $s_t = \|\hat{\mathbf{x}}_t - \mathbf{x}_t\|_2^2$, and flag $t$ as anomalous if $s_t$ exceeds a learned or adaptive threshold.

In this work, we focus on diffusion-based models, which corrupt the input with noise and then learn to reconstruct it, using the reconstruction error as the anomaly indicator.

## 4 METHODOLOGY

### 4.1 OVERALL FRAMEWORK

Figure 1 illustrates the architecture of our proposed framework. The model consists of two major components: (i) a *Spatio-Temporal Prior Module (STPM)* that generates smooth multi-resolution representations and similarity-guided graph priors, and (ii) a *Graph-Guided Reconstruction Diffusion model (GGRD)* that performs one-step reconstruction guided by these priors.

Given an input time series $\mathbf{X}$, the STPM first generates a set of multi-resolution series $\{\mathbf{X}^{(r)}\}_{r=1}^R$ via sliding-window averaging, where $R$ is the number of resolutions. For each resolution, we compute cosine similarity between feature dimensions to construct a similarity-guided graph tensor $\mathbf{S}^{(r)} \in \mathbb{R}^{P \times D \times D}$, which encodes the pairwise dependencies among features and time segments. The set $\{\mathbf{X}^{(r)}, \mathbf{S}^{(r)}\}$ is then corrupted by Gaussian noise following the forward diffusion process to obtain $\{\mathbf{X}_K^{(r)}\}$.

The GGN takes the noisy multi-resolution series as input and performs a *single-step* reconstruction to produce $\hat{\mathbf{X}}$. Unlike conventional diffusion models that require iterative denoising, GGN directly restores the clean signal in one step, significantly improving inference efficiency and mitigating error accumulation. The reconstruction is guided by graph priors through the *Graph-Guided Attention (GGA)* mechanism, enabling explicit modeling of dynamic cross-feature dependencies.

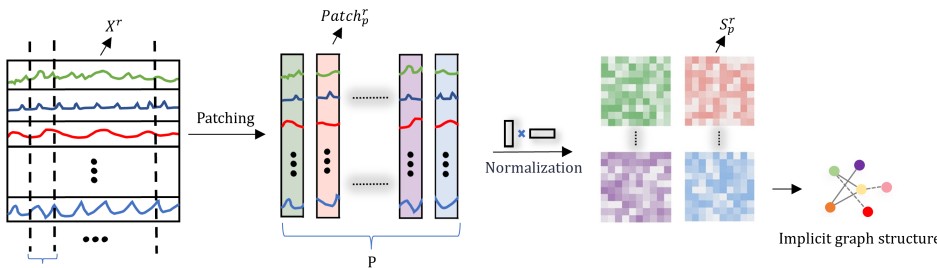

Figure 2: The process of obtaining similarity-guided Graph tensors.

## 4.2 SPATIO-TEMPORAL PRIOR MODULE

**Smooth multi-resolution decomposition**  To obtain multi-resolution representations, we apply a sliding window of length $rL$ with stride 1 over $\mathbf{X}$ at each resolution $r \in [1, R]$. For each window, we take the mean of the included points to form the smoothed sequence $\mathbf{X}^{(r)} \in \mathbb{R}^{T \times D}$. This overlapping-window design preserves temporal smoothness and continuity, mitigating the staircase artifacts introduced by non-overlapping pooling.

**Timestamp hard embedding**  Following D3R (Wang et al., 2023), we extract calendar-based features from absolute timestamps and build a fixed multi-granularity embedding $\mathbf{X}_{\text{time}} \in \mathbb{R}^{T \times d_{\text{time}}}$. We use $d_{\text{time}} = 5$ fields encoding *minute*, *hour*, *day*, *week*, and *month*. This hard-coded timestamp representation injects priors about periodic and seasonal patterns while preserving temporal ordering, and is subsequently consumed by the Time-Augment Encoder to enhance multi-level temporal modeling.

**Similarity-guided graph construction**  As in Figure 2, for each resolution $r$, we divide $\mathbf{X}^{(r)}$ into $P$ consecutive patches(time segments) along the temporal dimension:

$$P = \left\lceil \frac{T}{\text{patch size}} \right\rceil \tag{1}$$

For each patch $p$, we compute a cosine-similarity matrix:

$$\mathbf{S}_{ij}^{(r,p)} = \frac{\langle \mathbf{x}_{:,i}^{(r,p)}, \mathbf{x}_{:,j}^{(r,p)} \rangle}{\|\mathbf{x}_{:,i}^{(r,p)}\|_2 \, \|\mathbf{x}_{:,j}^{(r,p)}\|_2}, \tag{2}$$

where $\mathbf{x}_{:,i}^{(r,p)}$ denotes the series of feature $i$ in patch $p$. The collection $\{\mathbf{S}^{(r,p)}\}$ forms a dynamic graph prior that captures time-varying, multi-resolution feature dependencies.

## 4.3 DIFFUSION FORWARD PROCESS

Following denoising diffusion probabilistic models (DDPM) (Ho et al., 2020), we gradually inject Gaussian noise into each $\mathbf{X}^{(r)}$ over $K$ steps via a forward Markov chain:

$$q(\mathbf{X}_k^{(r)} \mid \mathbf{X}_{k-1}^{(r)}) = \mathcal{N}\big(\sqrt{1 - \beta_k}\,\mathbf{X}_{k-1}^{(r)}, \beta_k \mathbf{I}\big), \quad k = 1, \ldots, K, \tag{3}$$

where $\beta_k \in (0, 1)$ is a variance schedule. In practice, $\mathbf{X}_k^{(r)}$ can be sampled in closed form at any $k$, allowing efficient generation of the fully-noised sample $\mathbf{X}_K^{(r)}$ in one step.

## 4.4 GRAPH-GUIDED NETWORK (GGN)

The GGN serves as the backbone for one-step reconstruction. It operates in a coarse-to-fine manner over $R$ resolution levels and $R - 1$ steps. At step $m$, we first fuse the noisy input $\mathbf{X}_K^{(R-m)}$ with the previous reconstruction $\hat{\mathbf{Z}}^{(m-1)}$ through a channel-wise concatenation and a 1D projection layer:

$$\mathbf{Z}^{(m)} = \text{Proj}\big([\mathbf{X}_K^{(R-m)}; \hat{\mathbf{Z}}^{(m-1)}]\big), \quad \hat{\mathbf{Z}}^{(0)} \equiv \mathbf{X}_K^{(R)}, \quad m = 1, \ldots, R - 1. \tag{4}$$

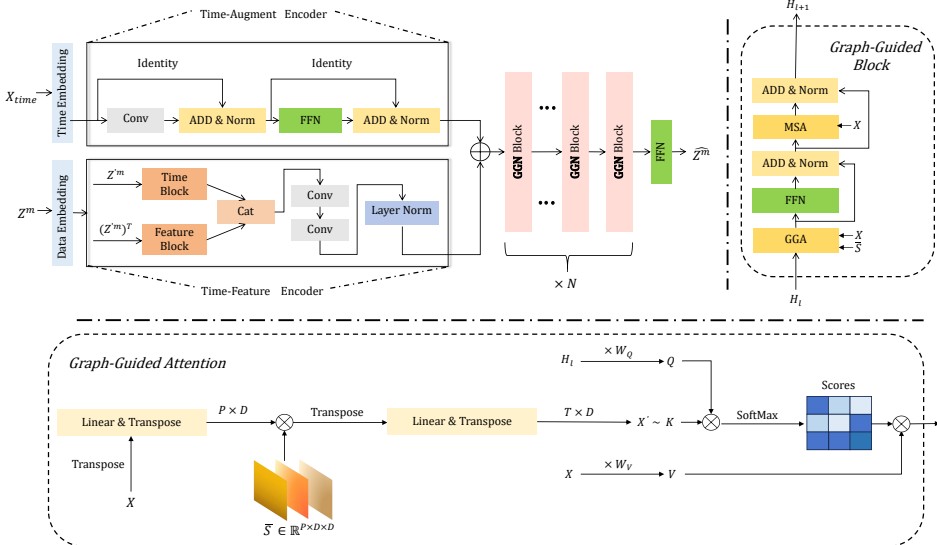

Figure 3: The overall structure of GGNetwork mainly includes Time-Feature Encoder, Time-Augment Encoder and some GGN Blocks. The structures of GGN Block and GGA are respectively to the right and below the dotted line.

This fusion passes information across resolutions, enabling progressively refined reconstruction.

**Dual encoders** The structure of GGN is presented in Figure 3.The fused sequence $\mathbf{Z}^{(m)}$ is embedded into a $d_{\text{model}}$-dimensional space and processed by two parallel encoders(the detailed structure is in the Appendix C.1): (i) the *Time-Feature Encoder* $\mathcal{E}_{\text{tf}}$, which extracts temporal dependencies and feature-wise interactions using self-attention and local convolutions, and (ii) the *Time-Augment Encoder* $\mathcal{E}_{\text{ta}}$, which projects the timestamp embedding $\mathbf{X}_{\text{time}}$ into the same space. Their outputs are added elementwise:

$$\mathbf{H}_0^{(m)} = \mathcal{E}_{\text{tf}}(\mathbf{Z}^{(m)}) + \mathcal{E}_{\text{ta}}(\mathbf{X}_{\text{time}}). \tag{5}$$

**Stacked GGN blocks with GGA** The hidden state is refined through $N$ stacked GGN blocks. Each block contains(as shown in Figure 3): (a) a **Graph-Guided Attention (GGA)** module that incorporates the similarity-guided graph prior $\bar{\mathbf{S}}^{(m)}$, which is obtained by a specially defined fusion function $f$ (see Appendix C.2 for details), representing the information fusion between $\mathbf{S}$ of the current step and $\mathbf{S}$ of the previous step,and (b) a temporal self-attention layer that maintains global temporal context.

Given queries $\mathbf{Q}$, values $\mathbf{V}$ projected from $\mathbf{H}_\ell^{(m)}$, Linear&T$(\cdot)$ represents passing through the linear layer and immediately transposing.GGA modifies the attention computation by applying the graph prior to the keys:

$$\mathbf{K} = \text{Linear\&T}_2\big(\text{Linear\&T}_1(\mathbf{X}^{\mathbf{T}}) \odot \bar{\mathbf{S}}^{(m)}\big) \tag{6}$$

$$\mathbf{A}^{(m)} = \text{softmax}\Big(\frac{\mathbf{Q}\mathbf{K}^{\top}}{\sqrt{d}}\Big), \qquad \text{GGA}(\mathbf{H}) = \mathbf{A}^{(m)}\mathbf{V}. \tag{7}$$

This biases attention weights toward feature pairs with higher similarity in $\bar{\mathbf{S}}^{(m)}$, explicitly capturing dynamic cross-feature dependencies. The GGA output is added to the temporal attention output and passed through a lightweight feed-forward layer with residual connections and layer normalization.

**Stage output** After $N$ blocks, the hidden representation is projected back to the input dimension to yield the reconstruction:

$$\hat{\mathbf{Z}}^{(m)} = \text{Head}(\mathbf{H}_N^{(m)}) \in \mathbb{R}^{T \times D}, \tag{8}$$

which is then passed to the next resolution level as input. The final reconstruction $\hat{\mathbf{X}} = \hat{\mathbf{Z}}^{(R-1)}$ is used for anomaly scoring.

### 4.5 TRAINING OBJECTIVE AND ANOMALY SCORING

We train GGRD to minimize the mean squared error (MSE) between the original and reconstructed series:

$$\mathcal{L} = \frac{1}{TD} \sum_{t=1}^{T} \sum_{d=1}^{D} \left( x_{t,d} - \hat{x}_{t,d} \right)^2. \tag{9}$$

During inference, the anomaly score at time step $t$ is defined as the reconstruction error $s_t = \|\hat{\mathbf{x}}_t - \mathbf{x}_t\|_2^2$. Anomalies are identified by applying an adaptive thresholding method SPOT (Siffer et al., 2017) to $\{s_t\}_{t=1}^{T}$.

## 5 EXPERIMENTS

### 5.1 EXPERIMENTAL SETTINGS

**Datasets**   This paper mainly uses five public time series datasets,including PSM (Pooled Server Metrics)(Abdulaal et al., 2021),SMD(Server Machine Dataset)(Su et al., 2019),and SWaT(Secure Water Treatment)(Mathur & Tippenhauer, 2016).The training set and validation do not contain labels,only the test set data has labels.More descriptions of the datasets are in Appendix A.1.

**Metrics**   The experiments adopt Precision, Recall and F1 score as the main evaluation metrics. Unlike the point adjustment method commonly used in most existing studies(Chen et al., 2023; Xiao et al., 2023; Wen et al., 2025), we use the Affiliation based(Huet et al., 2022) strategy to calculate the indicators. In a continuous anomaly interval, as long as any point is predicted, the point adjustment strategy will be regarded as the entire interval being detected, thereby significantly improving TP and masking the deficiency of the model in anomaly localization, which is prone to cause false performance improvement(Wang et al., 2023). In contrast, the method based on Affiliation measures the matching relationship between the predicted anomalies and the true anomaly intervals, which more objectively reflects the performance of the model in interval-level anomaly detection. It can avoid excessive bias towards long interval anomalies and thus obtain more reasonable results.

**Experiment setup**   GGRD uses Adam as the optimizer, with the learning rate set to $1e-4$ and weight decay set to $1e-4$. For the unlabeled data in each dataset, we select $80\%$ as the training set, $20\%$ as the validation set, and the labeled data as the test set. For all datasets, the size of batchsize is set to 8 and the training epoch is 10. Go through $1,000$ steps from $0.0001$ to $0.02$. The number of GGN blocks $N$ is set to 4 in the SMD dataset and 2 in the rest of the datasets. The number of noise additions $K$ to the original time series data is 500. All the experiments of GGRD were carried out under the Linux system, Pytorch, and a total of 4 NVidia A100 GPUs were used.

**Baseline**   The baseline methods we selected cover multiple paradigms, including probabilistic modeling, linear transformation, deep neural networks, and Transformer methods, etc. For specific descriptions, please refer to the Appendix A.2.

### 5.2 DETECTION RESULTS

We conducted experiments on the GGRD model and baselines on multiple real-world datasets, and the experimental results are shown in Table. 1. The GGRD model achieved the best performance on the non-stationary datasets PSM and SMD, which were $3.28\%$ higher than the second-best results (from $0.8100$ to $0.8428$) and $1.44\%$ higher (from $0.9238$ to $0.9382$), respectively. The mean of the overall F1 on all datasets, Avg-F1, was $0.2\%$ higher than the second-best result (from $0.8420$ to $0.8440$). However, on the SWaT dataset, GGRD performs slightly worse than D3R and MODEM. This is mainly because SWaT data usually contains short-term burst exceptions, while the modeling mechanisms of D3R and MODEM are more sensitive to such instantaneous changes. In contrast, the advantage of GGRD lies in its ability to effectively capture the dependencies between non-stationary features and complex features. Therefore, on time series datasets such as PSM and SMD, which have complex dynamic behaviors and cross-feature correlations, GGRD demonstrates more robust and superior anomaly detection performance, but it is not the case in SWaT(see the details in Appendix A.4). These results indicate that although there is a slight gap in specific short-term sudden abnormal scenarios, GGRD still has significant advantages when dealing with real complex environments and high-dimensional multi-variable data.

Table 1: The comparison test results with other models on three real-world datasets show that our GGRD leads on most datasets. The best F1 score is marked in bold, and the second best is marked with an underline. Avg-F1 represents the average F1 score.

| Method | PSM | | | SMD | | | SWaT | | | Avg-F1 |
|---|---|---|---|---|---|---|---|---|---|---|
| | P | R | F1 | P | R | F1 | P | R | F1 | |
| COPOD | 0.7602 | 0.3175 | 0.4479 | 0.6676 | 0.1366 | 0.2268 | 0.9876 | 0.1180 | 0.2108 | 0.2952 |
| ECOD | 0.7460 | 0.3384 | 0.4656 | 0.7398 | 0.1615 | 0.2651 | 0.9761 | 0.1151 | 0.2059 | 0.3122 |
| OCSVM | 0.8761 | 0.4744 | 0.6155 | 0.0000 | 0.0000 | 0.0000 | 0.6196 | 0.7558 | 0.6810 | 0.4322 |
| CBLOF | 0.5990 | 0.9845 | 0.7449 | 0.8667 | 0.3352 | 0.4834 | 0.6308 | 0.7091 | 0.6677 | 0.6320 |
| HBOS | 1.0000 | 0.0654 | 0.1228 | 0.5628 | 0.8007 | 0.6610 | 0.5771 | 0.8049 | 0.6722 | 0.4853 |
| IForest | 1.0000 | 0.0335 | 0.0648 | 1.0000 | 0.0937 | 0.1713 | 0.6127 | 0.6280 | 0.6203 | 0.2855 |
| LODA | 0.9266 | 0.4017 | 0.5605 | 0.5902 | 0.6618 | 0.6240 | 0.6117 | 0.7014 | 0.6535 | 0.6127 |
| VAE | 0.6221 | 0.8772 | 0.7280 | 0.8209 | 0.4349 | 0.5686 | 0.6355 | 0.7218 | 0.6759 | 0.6575 |
| DeepSVDD | 0.7405 | 0.5064 | 0.6015 | 0.6498 | 0.6477 | 0.6488 | 0.5911 | 0.9353 | 0.7244 | 0.6582 |
| LSTM-AE | 0.7511 | 0.7586 | 0.7548 | 0.8496 | 0.4349 | 0.5753 | 0.6018 | 0.7219 | 0.6564 | 0.6622 |
| MTAD-GAT | 0.7990 | 0.6014 | 0.6863 | 0.8590 | 0.6769 | 0.7571 | 0.6590 | 0.7751 | 0.7123 | 0.7186 |
| TFAD | 0.7914 | 0.7163 | 0.7520 | 0.5632 | 0.9783 | 0.7149 | 0.6038 | 0.8196 | 0.6953 | 0.7207 |
| Anomaly Transformer | 0.5201 | 0.8504 | 0.6455 | 1.0000 | 0.0319 | 0.0619 | 0.5541 | 0.5994 | 0.5759 | 0.4278 |
| Diff-AD | 0.5564 | 0.7674 | 0.6450 | 0.5014 | 0.9093 | 0.6464 | 0.5183 | 0.7979 | 0.6284 | 0.6399 |
| D3R | 0.6294 | 0.9619 | 0.7609 | 0.7715 | 0.9926 | 0.8682 | 0.7206 | 0.8529 | 0.7812 | 0.8034 |
| Imdiffusion | 0.7556 | 0.8784 | 0.8100 | 0.9605 | 0.5271 | 0.6741 | 0.8387 | 0.2058 | 0.3297 | 0.6046 |
| MODEM | 0.7348 | 0.8755 | 0.7990 | 0.8918 | 0.9582 | 0.9238 | 0.7436 | 0.8732 | **0.8032** | 0.8420 |
| **ours** | 0.8827 | 0.8064 | **0.8428** | 0.9812 | 0.8988 | **0.9382** | 0.6529 | 0.8840 | 0.7511 | **0.8440** |

## 5.3 ABLATION STUDIES

To verify the role of each module in the model, we conducted systematic ablation experiments on multiple datasets and compared the results with those of the complete model,and the ablation results are shown in Table 2.

Table 2: The ablation experiment results reported the best F1.

| Dataset | GGRD | w/o time-feature | w/o time-augment | w/o gga | w/o sliding window | w/o timestamp |
|---|---|---|---|---|---|---|
| SMD | **0.9382** | 0.8922 | 0.9271 | 0.8515 | 0.9181 | 0.8931 |
| PSM | **0.8428** | 0.8078 | 0.8168 | 0.7721 | 0.8089 | 0.8231 |
| SWaT | **0.7511** | 0.7267 | 0.7331 | 0.7117 | 0.7377 | 0.7404 |
| average | **0.8440** | 0.8089 | 0.8257 | 0.7784 | 0.8216 | 0.8189 |

**Spatio-Temporal Prior Module** This module is mainly used to generate hard-coded timestamps, multi-resolution data and graph structures. In the ablation experiment, we removed the timestamps, denoted as **w/o timestamp**, and replaced the generation method of multi-resolution data from the proposed sliding window to non-overlapping average pooling(Zhong et al., 2025), denoted as **w/o sliding window**. The experimental results in Table 2 show that after removing the timestamp on the SMD dataset, the F1 value decreased from $0.9382$ to $0.8931$, and the performance decreased by $4.51\%$. Similarly, after replacing the sliding window with non-overlapping average pooling, the F1 of all three datasets decreased, indicating that timestamps and efficient multi-resolution modeling methods are of great significance for capturing the dynamic patterns of time series.

**Reconstruction of Multivariate Time Series Module** This module is the core part of the GGRD proposed in this paper, including Time-Feature Encoder, Time-Augment Encoder and GGA. We remove one of the components respectively, denoted as **w/o time-feature, w/o time-augment and w/o gga**. The experimental results show that on multiple datasets, the F1 value of the complete model is always superior to that of the ablated version. For example, in PSM, the F1 value of the original model was $0.8428$, but it decreased to $0.8078$ after removing the time-feature, a reduction of $3.5\%$. In SMD, the best F1 value decreased from $0.9382$ to $0.8515$ after removing GGA, with the largest performance decline and other datasets also have a significant impact, further verifying the key role of GGA in capturing cross-dimensional dependencies and dynamic features.

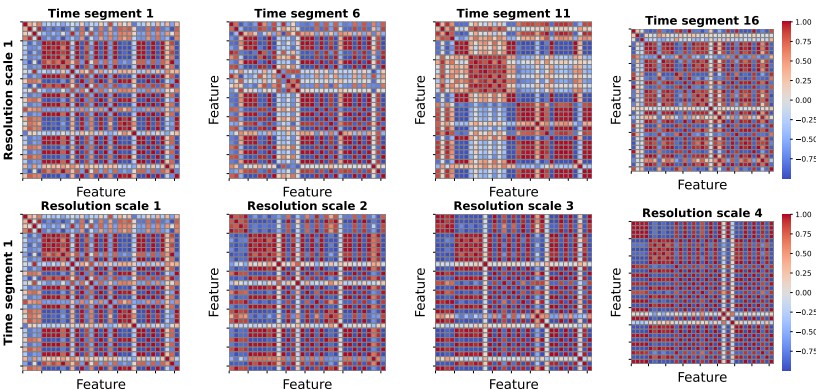

Figure 4: Visualization of Similarity-guided Graph Tensor **S** on SMD. The four matrices in the first row represent the changes at different time segments when the resolution scale is 1. The second row represents the changes in different resolution scales within the same time segment (time segment 1).

## 5.4 EFFECTIVENESS ANALYSIS

**Similarity-Guided Graph Tensor** To further illustrate the effectiveness of GGA, in the previous paper, Similarity-guided Graph Tensor **S** was constructed based on cosine Similarity, and the **S** was visualized at four resolution scales and any four time segments of the SMD dataset. As shown in the Figure 4, the similarity between features shows significant differences at different time segments and resolutions, indicating that the feature dependency relationship is dynamic and multi-scale. For instance, when the resolution scale is 1(the first row), the similarity between features at different time segments varies the most. Similarly, when the time segmen is fixed (the second row), the feature similarity also shows significant differences at different resolutions.Compared with the methods that do not explicitly consider this information, **S** can provide additional correlation information for multi-head self-attention, thereby enabling GGA to better capture cross-dimensional dependencies and improve the expressive ability of the model in anomaly detection.

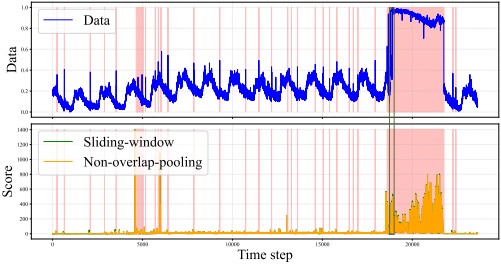 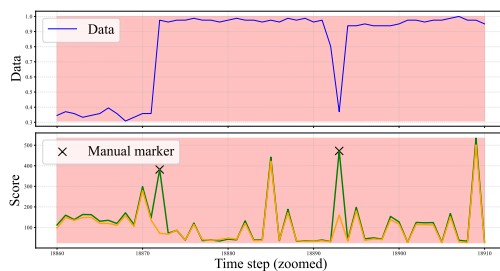

(a)Real data and abnormal scores.      (b)An example.The details in the green box.

Figure 5: Real data and the abnormal scores corresponding to the two methods respectively. Both methods obtain abnormal scores under their own optimal models. The higher the abnormal score, the easier it is to be detected as an anomaly. The two types of data show differences at non-stationary points.

**Obtain multi-resolution data using sliding windows** When initially constructing the sliding window, we reduce the information loss between adjacent times. We use overlapping sliding Windows instead of non-overlapping average pooling. In Figure 5(a), the upper part is the original data of a feature in the SMD dataset, and the red transparent background represents the anomaly. The lower part shows the magnitudes of the anomaly scores corresponding to the two data generation methods. And the Figure 5(b) is an example. The black manually marked areas show obvious differences

because the real data of the adjacent timestamps corresponding to the two places are changing drastically. The sliding window method, being smoother compared to the latter method and less likely to lose more information, can obtain higher outliers, while lower outliers may lead to missed detections of anomalies. Similarly, when the label is normal, non-overlapping methods that are not stationary may also receive higher scores and be misjudged as abnormal.

### 5.5 HYPERPARAMETER ANALYSIS

**Hyperparameters of GGRD**   Here we mainly investigated the influence of the number of resolution categories $R$, the number of GGN modules $N$, $patch\ size$, and the initial window length $L$ on the model performance.The specific experimental results can be found in the Appendix A.3.

**Anomaly detection threshold**   Given a probability $q$, the SPOT algorithm can automatically obtain the detection threshold by using the abnormal scores of the training data and the test set. The following figure presents the metrics of different datasets under different $q$. In this experiment, we evaluated using the SPOT algorithm on three datasets (SWaT, PSM, and SMD), and observed the changes in Precision, Recall, and F1-score by adjusting the threshold $q$. The results are presented in Figure 6. The results show that as the $q$ value increases, it is often accompanied by an increase in Recall, and Precision usually decreases, and $q$ represents the proportion of outliers among extreme points(Siffer et al., 2017). This trend is particularly evident in the SWAT and PSM datasets, with the optimal F1-scores appearing at $q = 0.007$ and $q = 0.02$, respectively. On the SMD dataset, the model as a whole demonstrates high stability and robustness, with relatively stable Precision and Recall. Therefore, it is necessary to seek a trade-off between Precision and Recall to achieve the best F1. Overall, the SPOT algorithm can achieve high anomaly recognition results on different types of time series data and strike a good balance between accuracy and recall by reasonably selecting $q$, demonstrating its applicability and reliability in multiple scenarios.

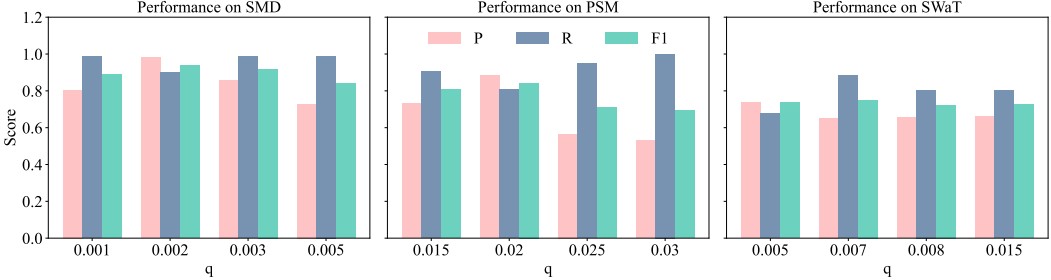

Figure 6: The influence of different $q$ on Model performance

## 6 CONCLUSION AND LIMITATION

**Conclusion.** This paper proposes a **G**raph **G**uided **R**econstruction **D**iffusion Model (GGRD) to address the deficiencies of existing methods in anomaly detection of multivariate time series. Specifically, we generate multi-resolution data through a sliding window, reducing the resolution while retaining the original information features as much as possible. Furthermore, in this study, the reconstruction structure of GGRD was carefully designed and the correlation between features was considered. Similarity-Guided Graph Tensors were introduced into GGN to effectively guide feature interaction and improve the accuracy and robustness of data modeling. A large number of experiments have shown that GGRD outperforms existing anomaly detection methods on various datasets.

**Limitations.** The length of the time segment in GGRD is fixed (depending on the $patch\ size$ and $T$), but in fact, the correlation between time series features does not remain constant over a fixed-length time segment. That is to say, the duration of a certain correlation situation varies and is often highly uncertain. Therefore, a method for dynamically obtaining the graph structure prior is needed to enhance the detection ability of irregular fluctuations and short-term sudden anomalies. At the same time, in addition to cosine similarity (even though it is simple and effective), other forms can be considered for the graph structure tensor to further improve the modeling ability of time series.

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

# A  EXPERIMENTAL DETAILS

## A.1  DATASETS

Just like D3R(Wang et al., 2023),we only retained the continuous variables in the data for the experiment and provided the statistical data of these datasets in the Table 3.

Table 3: Dataset statistical description(AR represents the Rate of Anomalies in the test set).

| Dataset | Training size | Testing size | Dimensions | Frequency | AR(%) |
|---------|---------------|--------------|------------|-----------|-------|
| SMD | 23688 | 23689 | 33 | 1 minute | 15.7 |
| PSM | 132481 | 87841 | 25 | 1 minute | 27.8 |
| SWAT | 6840 | 7500 | 25 | 1 minute | 12.6 |

## A.2  BASELINE

Some of the baselines selected for the experiment adopted the results of D3R and were implemented following the configuration recommended in the original paper. Apart from the first one, the other baselines are based on our operation and run with the recommended configuration in the original paper.

**COPOD**(Li et al., 2020)    COPOD innovatively utilizes copula to construct an empirical distribution and calculate the tail probability, thereby achieving a parameterless, interpretable and efficient anomaly detection method.

**ECOD**(Li et al., 2022)    Based on the empirical cumulative distribution function to estimate the tail probability, a parameter-free, easily interpretable and highly efficient and scalable anomaly detection method has been implemented, which significantly outperforms existing methods on large-scale and high-dimensional data.

**OCSVM**(Schölkopf et al., 2001)    This method extends the support vector machine to unsupervised scenarios, constructs a discriminant function through the kernel function, divides the input space into high-probability subsets and their complements to ensure that the probability of new samples falling into this subset is controlled, and solves the extended coefficients through quadratic programming to achieve efficient anomaly detection.

**CBLOF**(He et al., 2003)    It is an unsupervised anomaly detection method based on local clusters, which identifies anomalies by evaluating the behavioral significance of data points in their respective clusters and can effectively discover outliers with physical or statistical significance.

**HBOS**(Goldstein & Dengel, 2012)    It is an unsupervised anomaly detection method based on histograms. By assuming feature independence, it achieves linear time scoring and can efficiently identify global anomalies, but it performs weakly in local anomaly detection.

**IForest**(Liu et al., 2008)    Unsupervised anomaly detection methods based on the idea of isolation achieve linear time complexity and low memory consumption by explicitly isolating outliers and using sub-sampling, and exhibit excellent performance on large-scale, high-dimensional or datasets with irrelevant features.

**LODA**(Pevnỳ, 2016)    LODA is an unsupervised anomaly detection method based on weak detector integration, which can efficiently handle large-scale or streaming data, deal with missing variables and concept drift, and simultaneously identify the characteristics of anomaly occurrence. It outperforms many existing methods in terms of speed and accuracy.

**VAE**(Kingma & Welling, 2013)    By reparameterizing the variational lower bound and approximating the model, efficient learning and inference of directed probabilistic models with continuous latent variables have been achieved, and effective optimization can be carried out even in the case of posteriorly unresolvable and large-scale datasets.

**DeepSVDD**(Ruff et al., 2018)  It is a deep method trained with anomaly detection as the goal. By directly optimizing the anomaly detection target in the neural network, it shows good results in image benchmark datasets and adversarial sample detection.

**LSTM-AE**(Kieu et al., 2018)  This model generates statistical features for time series and reconstructs them using autoencoders to capture representative patterns, thereby identifying outliers that deviate from the reconstruction. At the same time, it combines convolutional and LSTM networks as well as context information to improve the accuracy of anomaly detection.

**MTAD-GAT**(Zhao et al., 2020)  By capturing the dependencies of multivariate time series in the time and feature dimensions through the parallel graph attention layer, and combining prediction and reconstruction optimization, efficient anomaly detection is achieved, while also having good interpretability and anomaly diagnosis capabilities.

**TFAD**(Zhang et al., 2022)  TFAD utilizes time-frequency joint analysis and enhances anomaly detection performance and interpretability through time series decomposition and data augmentation mechanisms.

**Anomaly Transformer**(Xu et al., 2021)  By using the self-attention mechanism to calculate the correlation differences of points in the time series, and amplifying the distinguisability between normal and abnormal through abnormal attention and the minim-to-maximum strategy, advanced performance has been achieved in various unsupervised time series anomaly detection tasks.

**Diff-AD**(Xiao et al., 2023)  By using the density ratio to select normal observations and combining with denoising diffusion interpolation with increasing conditional weights and multi-scale state space modeling, abnormal concentration scenarios can be effectively handled, achieving stable multi-variable time series anomaly detection.

**D3R**(Wang et al., 2023)  By combining the dynamic decomposition of data-time hybrid attention with noise diffusion reconstruction, the time series is split into stable components and trend components, and the non-stationary multivariable time series is processed end-to-end, significantly reducing the false alarm rate caused by drift and improving the detection performance.

**Imdiffusion**(Chen et al., 2023)  Combining time series interpolation with diffusion models, anomaly prediction is carried out through stepwise denoising to generate signals, accurately modeling time series and cross-variable dependencies.

**MODEM**(Zhong et al., 2025)  By jointly modeling non-stationary time series through multi-resolution diffusion and frequency-domain augmented networks, cross-resolution correlations are captured during the coarse-to-fine generation process to achieve precise anomaly detection of complex time series patterns.

## A.3 HYPERPARAMETER OF GGRD ANALYSIS

The results are shown in Figure 7, and the following conclusions can be drawn: (1) As $R$ increases, performance shows certain differences on different datasets. Overall, better results are usually achieved when $R = 4$. For example, on the SMD dataset, the F1-score reached 0.9382 when $R = 4$, and the performance declined when it increased further. The lower resolution ($R = 2$) leads to insufficient expressive power of the model and a decline in discrimination. However, excessive resolution division can lead to overly fine division, which affects the model's judgment of the overall trend of the time series and instead impacts the model's discriminative ability. (2) The effect of increasing the number of GGN blocks on different datasets is not consistent. On the SMD dataset, the performance is the best when $N = 4$, while on PSM and SWaT, too many GGN blocks ($N = 6, 8$) instead lead to a decline in performance. This indicates that a balance needs to be struck between complexity and expressive power, and overly deep stacking is not always beneficial. (3) Appropriate patch partitioning affects the shape of $\mathbf{S}$ and alters the number of time segmens a time series is divided into. The results show that on each dataset, a smaller $patch\ size$ (such as 4 or 6) can achieve a higher F1-score, while an overly large size (such as 8) will lead to information loss. It is indicated that overly large patches will weaken the ability to capture local patterns. (4) The initial length $L$, in coordination with $R$, determines the size of the sliding window. It can be found that in any dataset, the smaller $L$ can achieve better results.

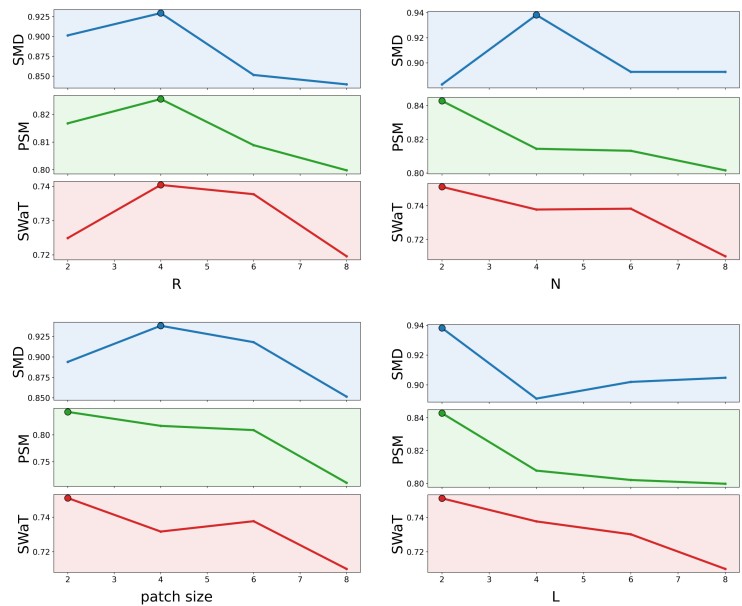

Figure 7: Parameter sensitivity analysis on $R$, $N$, $patch\ size$ and $L$

Based on these analyses, the optimal parameters of the SMD, PSM and SWaT datasets on $R$,$N$,$patch\ size$ and $L$ are respectively $\{4, 4, 4, 2\}$,$\{4, 2, 2, 2\}$, and $\{4, 2, 2, 2\}$.

### A.4 VISUALIZATION OF THE PROPERTIES OF SWAT DATASETS

The main advantage of GGRD lies in its ability to capture the similarity between displayed features. As shown in Figure 9, the values of **S** in the SWaT dataset do not show significant differences at different time periods (Time segment 1, 11, 21, 32) and different resolution scales (Resolution 1, 2,3,4). This also limits the modeling upper limit of GGRD to a certain extent. Compared with large and cumbersome models like MODEM, the performance of our GGRD on the SWaT dataset is slightly lower.

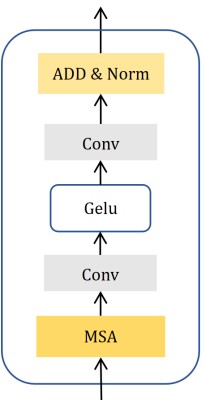

Figure 8: Time Block and Feature Block.

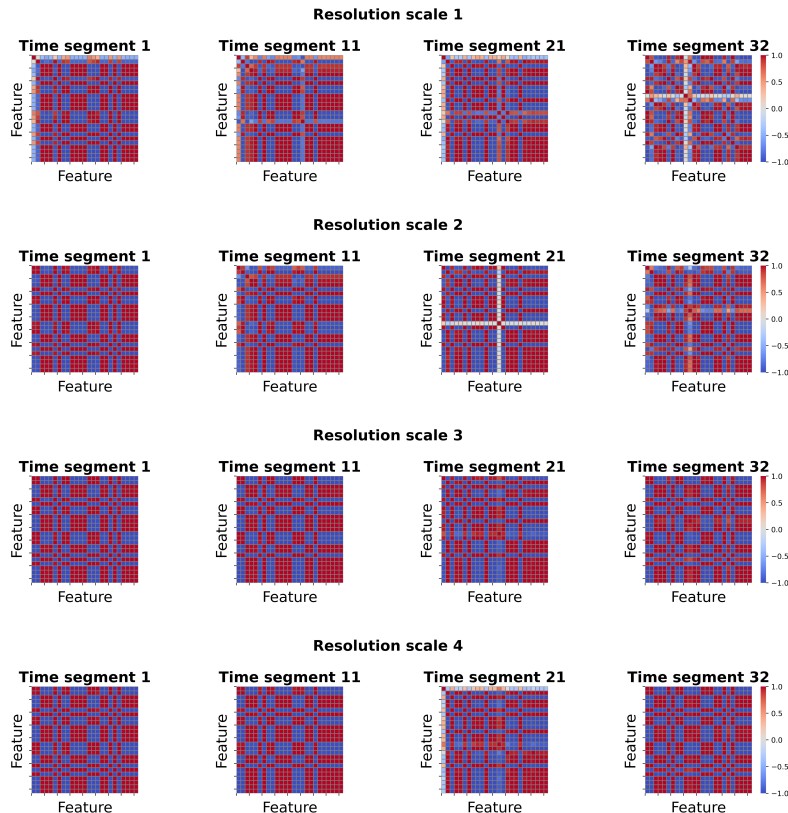

Figure 9: Visualization of Similarity-guided Graph Tensor **S** on SWaT.

## B   ADDITIONAL EXPERIMENTS

### B.1   ONE STEP VS MULTI-STEP

The core advantage of multi-step iterative denoising (such as DDPM) lies in generating high-quality and high-fidelity samples, which is crucial in tasks like image generation as it requires creating details from scratch.

However, in anomaly detection based on refactoring, our goal is not to 'create', but to 'test'. The ultimate goal is to calculate the reconstruction error. To verify the efficiency advantage of one-step denoising, we designed a multi-step denoising experiment. In multi-step denoising training, the model learns to recover from any noise level $X_k$ to $X_{k-1}$, and in the inference stage, it iterates step by step from high noise to the final output in a completely consistent manner. We evaluated the impact of different denoising steps on model performance and computational efficiency, and the experiment was independently repeated five times.

The results in Table. 4 show that a more detailed multi-step denoising setting does not lead to an improvement in performance. Instead, it results in a significant increase in training and inference time. This further demonstrates the superiority of one-step denoising Settings in reconstruction-based anomaly detection tasks.

### B.2   THE INFLUENCE OF NOISE ADDITION FREQUENCY

To illustrate the necessity and effectiveness of the denoising process of diffusion in this time series anomaly detection task, we conducted comparative experiments on $K$ under different datasets and compared the classification metrics and MSE on the test sets. It can be found that the impact of the denoising level on the anomaly detection performance is significant, which proves the necessity of denoising in the model.

Table 4: The performance and efficiency of the model under different denoising steps on the SWaT dataset, with the best results indicated in bold.

| Denoising Steps | F1-score (mean $\pm$ std) | Training Time (min) | Inference Time (min) |
|---|---|---|---|
| 1-step | **0.7411 $\pm$ 0.0105** | **27.40** | **5.84** |
| 5-steps | 0.6855 $\pm$ 0.0185 | 48.33 | 28.16 |
| 10-steps | 0.6748 $\pm$ 0.0159 | 54.01 | 39.23 |

The experimental results Table. 5 indicate that either no noise ($K = 0$) or insufficient noise will affect the discriminative ability of anomaly detection. When the noise is moderate ($K = 500$), all the indicators of the model reach the optimum. Meanwhile, the results of MSE also show that in the task of time series anomaly detection based on reconstruction, a smaller reconstruction error does not necessarily lead to a better detection effect. In the reconstruction model, the model reconstructs by learning normal patterns to identify outliers. If no noise is added or the noise is insufficient during training or inference, the model may overfit the training data. It reconstructs the training set well but has poor generalization ability, resulting in significant reconstruction errors for normal samples with slight changes in the test set or actual scenarios, which affects abnormal judgment. Adding noise can force the model to learn denoising ability, enabling it to correctly reconstruct the normal mode even when the input contains disturbances, thereby enhancing the robustness of the reconstruction and the discrimination ability of anomaly detection. Appropriately increasing the number of noisy steps $K$ can make the model pay more attention to the overall pattern of the sequence rather than point-by-point fitting, causing more obvious reconstruction differences in outliers and thereby improving the performance of anomaly detection.

Table 5: The influence of different noise levels $K$ on the performance of anomaly detection, with the minimum value is bolded.

| Datasets | $K$ | P | R | F1 | MSE |
|---|---|---|---|---|---|
| SMD | 0 | 0.8159 | 0.9537 | 0.8794 | 15.64 |
| | 100 | 0.8088 | 0.8620 | 0.8346 | 15.85 |
| | 300 | 0.8491 | 0.9621 | 0.9021 | 16.98 |
| | 500 | 0.9812 | 0.8988 | **0.9382** | **14.74** |
| PSM | 0 | 0.7497 | 0.8368 | 0.7909 | 0.2540 |
| | 100 | 0.7160 | 0.8999 | 0.7975 | **0.2397** |
| | 300 | 0.7991 | 0.7943 | 0.7967 | 0.2563 |
| | 500 | 0.8827 | 0.8064 | **0.8428** | 0.2912 |
| SWaT | 0 | 0.5657 | 0.8889 | 0.6914 | 318.84 |
| | 100 | 0.6836 | 0.7665 | 0.7227 | **318.80** |
| | 300 | 0.5748 | 0.8887 | 0.6981 | 318.93 |
| | 500 | 0.6529 | 0.8840 | **0.7511** | 319.02 |

### B.3 REPEATED EXPERIMENT STATISTICS

To verify the stability of the model's performance. We independently conducted five experiments to obtain the standard deviation of F1 and the 95% confidence interval. As shown in Table. 6, the standard deviation is controlled within 0.01, and the model demonstrates high stability and robustness on each dataset.

### B.4 COMPARED WITH THE OVERHEAD OF OTHER ADVANCED MODELS

We compared the model overhead with the existing advanced models MODEM and D3R. The results of Table. 7 show that the proposed GGRD model has relatively reasonable training and inference overhead while ensuring detection performance. Compared with D3R, the total number of parame-

Table 6: Repeated Measures Experiment

| Datasets | F1-score (mean $\pm$ std) | 95% CI (t-method) |
|---|---|---|
| PSM | $0.8409 \pm 0.0171$ | [0.8137, 0.8682] |
| SMD | $0.9264 \pm 0.0137$ | [0.9045, 0.9483] |
| SWaT | $0.7411 \pm 0.0105$ | [0.7281, 0.7542] |

ters of GGRD has decreased by half, and the inference time is approximately 10 minutes, which is acceptable for actual deployment and maintenance.

Table 7: The comparison with D3R regarding training time, inference time and parameters, with the best ones in bold

| Model | Training Time (min) | Inference Time (min) | Total Params (MB) |
|---|---|---|---|
| D3R | 42.33 | 18.29 | 199.18 |
| MODEM | 38.48 | 14.12 | **34.28** |
| GGRD (ours) | **28.70** | **10.31** | 84.92 |

### B.5 REGARDING THE RELATIONSHIP BETWEEN PATCH SIZE AND EFFICIENCY

(i) The specific meaning of the 'graph structure' we proposed is the correlation between the features of different sensors within different time periods. Nodes represent different features, while edges represent the relationships between features. Given a $\mathbf{X}^{(\mathbf{r})} \in \mathbb{R}^{T \times D}$, we obtain the graph structure S by partitioning $patch$. Therefore, the size of $patch\ size$ will affect the size of P, but will not influence the inherent number of nodes $D$. $P$ represents the number of time periods that divide the time series of a sliding window $T$ into.

(ii) Regarding the efficiency issue in handling long time series. Our model does not handle the entire infinitely long time series at one time. In practical applications, we perform the calculation within a fixed-length sliding window ($T = 64, 128, 512$). This means that no matter how long the total length of the input time series $L$ is (weeks, months or even years), the computational complexity and memory consumption of our model at each time step are only related to the fixed window size $T$, and not to the total length $L$. The time complexity of constructing $S$ is $O(T \times D^2)$, which is independent of $patch\ size$. When GGA subsequently utilizes S, it only involves an additional matrix multiplication and does not change the order of complexity.

(iii) We verified the model training and inference efficiency under different $patch\ size$ and $T$ (based on the SWaT dataset) in Table. 8 and Table. 9, it can be found that as the $patch\ size$ decreases, the training and inference time does indeed increase. But the growth rate shows a clear nonlinear trend and does not increase exponentially. Therefore, a smaller $patch\ size$ does not significantly reduce time efficiency, which is determined by the complexity of our algorithm.

Table 8: Training and inference time under different patch sizes (**T = 64**).

| $patch\ size$ | **Training Time (s)** | **Inference Time (s)** |
|---|---|---|
| 2 | 1644.25 | 350.62 |
| 4 | 1367.52 | 273.94 |
| 8 | 1221.85 | 232.58 |
| 16 | 1136.84 | 215.79 |
| 32 | 703.78 | 124.81 |

Table 9: Training and inference time under different patch sizes (**T = 256**).

| patch size | Training Time (s) | Inference Time (s) |
|:---:|:---:|:---:|
| 2 | 4582.70 | 1128.08 |
| 4 | 2668.35 | 700.57 |
| 8 | 2430.75 | 550.15 |
| 16 | 1987.99 | 526.45 |
| 32 | 2153.20 | 505.33 |

## C  SOME SPECIFIC MODEL COMPONENTS

### C.1  TIME BLOCK AND FEATURE BLOCK STRUCTURE

The structures of the Time Block and the Feature Block are shown in the Figure 8, except that the shapes of the input data are different (non-transposed and transposed). Firstly, the input data is exchanged for information through the multi-head self-attention (MSA) module and processed by the GELU activation function. Subsequently, local features are extracted through the convolutional layer and the final output is obtained by connecting the residual and normalizing the layer. This Block is designed to extract the time information and feature information from the data respectively, thereby providing an effective representation for subsequent time series modeling and feature interaction.

### C.2  FUSION FUNCTION

In the process of coarse-to-fine reconstruction of time series data, for step $m$, in order to obtain the similarity matrix $\bar{\mathbf{S}}^{(m)}$ and integrate the previous similarity information and also take into account the current one, a function $f$ is defined:

$$\bar{\mathbf{S}}^{(m)} = f(m), \tag{10}$$

$$f(m) = \frac{1}{2}\left(\frac{1}{m}\sum_{i=R-m+1}^{R}\mathbf{S}^{(i)} + \mathbf{S}^{(R-m)}\right)(m>1), \quad f(1) = \frac{\mathbf{S}^{(R)} + \mathbf{S}^{(R-1)}}{2}. \tag{11}$$

## D  LLM USAGE

We only use LLM for aid or polish writing in a very small part of the article.

