# OpenReview forum: "Graph-Guided Reconstruction Diffusion for Multivariate Time Series Anomaly Detection"
_ICLR.cc/2026/Conference — ICLR 2026 Conference Withdrawn Submission_

### Official Review · Reviewer_r8wz · 2025-10-18

**Soundness:** 1
**Presentation:** 1
**Contribution:** 2
**Rating:** 2
**Confidence:** 5

**Summary:**

This paper proposes a time series anomaly detection method that integrates diffusion models with a graph attention mechanism. Specifically, the authors perform a horizontal multi-scale segmentation with noise injection and a vertical patch-based segmentation followed by cosine similarity computation. The resulting similarity matrix is treated as a graph, where each patch serves as a node, and the graph attention network is employed to reconstruct the denoised sequence. Finally, the proposed framework is evaluated on multiple real-world datasets, demonstrating its effectiveness.

**Strengths:**

1. The authors have provided their code, which improves the reproducibility and transparency of the study.
2. The proposed method achieves top-tier performance across multiple real-world datasets, demonstrating good empirical effectiveness.
3. Ablation studies are provided, validating the contribution of each component in the framework.

**Weaknesses:**

1. **Unclear story and motivation.**
   The manuscript fails to present a coherent narrative: it is not clear what concrete problem the authors are solving and why the proposed design choices (e.g., multi-resolution segmentation and similarity injection) are necessary. This issue starts in the *Introduction* and permeates the whole paper. In the *Methods* section, the presentation reads like a collection of disconnected subsections rather than a logically flowing design—readers are left uncertain how components interact and why each one is required.

2. **Limited novelty — component stacking.**
   The proposed pipeline largely consists of assembling existing components (multi-resolution decomposition, timestamp/hard embeddings, and similarity-graph construction [1-3]) without introducing fundamentally new algorithmic ideas. Worse, these components are treated almost independently and run in parallel, which weakens the claim of a unified, principled method.

3. **Problematic use of a “diffusion” framework.**
   The paper claims to be diffusion-based but abandons the core iterative SDE/ODE denoising paradigm by asserting one-step reconstruction from a noised sequence to a real sample. While such a shortcut may be used during training [4], the literature and theory suggest that iterative denoising is required at test time to obtain good reconstructions—especially when starting from high-noise or near-pure-noise states. The current presentation glosses over this mismatch between training and inference, undermining the validity of calling the method a diffusion framework.

4. **Scalability and computational cost of graph-guided design.**
   The similarity-guided graph construction appears to rely on small patch sizes for good performance (see Appendix Fig. 7). Small patches imply many graph nodes as sequence length grows, causing an increase in computation and memory. Given convergence and complexity concerns, I am not convinced this design is a generally viable optimization for long-term time series.

5. **Experimental issues and insufficient analysis.**
   (a) The use of point-adjusted F1 (F1-PA) is problematic because F1-PA can be inflated by trivial strategies and is regarded by parts of the community as an overly permissive metric. Relying on F1-PA without complementary, stricter evaluation undermines the experimental claims.
   (b) The “Visualization of Similarity-guided Graph Tensor” is not well motivated—it is unclear what this visualization is intended to demonstrate and whether the similarity tensor itself is learnable or fixed. Such analysis would be better placed in the motivation or method section, accompanied by a discussion of what the observed patterns imply for detection behavior.

6. **Writing quality.**
   The manuscript contains multiple grammatical and typographical errors throughout. The paper would benefit from a careful language and copy edit to improve clarity.

**Reference:**
[1] Zhong, Guojin, et al. "Multi-resolution decomposable diffusion model for non-stationary time series anomaly detection." The Thirteenth International Conference on Learning Representations. 2025.
[2] Wang, Chengsen, et al. "Drift doesn't matter: Dynamic decomposition with diffusion reconstruction for unstable multivariate time series anomaly detection." Advances in neural information processing systems 36 (2023): 10758-10774.
[3] Shen, Lifeng, Weiyu Chen, and James Kwok. "Multi-resolution diffusion models for time series forecasting." The Twelfth International Conference on Learning Representations. 2024.
[4] Yuan, Xinyu, and Yan Qiao. "Diffusion-TS: Interpretable Diffusion for General Time Series Generation." The Twelfth International Conference on Learning Representations.

**Questions:**

See the Weaknesses.

---

> ### Author Response · Authors · 2025-11-22
> **(Reply 1)Thank you for your valuable suggestions on our work!**
>
> Thank you for your valuable suggestions on our work. Regarding your question, we will answer it from the following aspects:
>
> ---
>
> 1. **Regarding the motivation of the article**
>
> The motivation for our work stems from a core challenge: how to effectively detect anomalies in a complex multivariate time series that combines non-stationarity, multi-scale patterns, and dynamic cross-variable dependencies. Our introduction clearly points out three fundamental limitations of existing methods in addressing this challenge.
>
> * **(i) Non-stationarity across resolutions:** Existing methods (such as modems) result in information loss and artifacts due to the use of non-overlapping pooling. For this purpose, we have introduced a smooth multi-resolution decomposition achieved through overlapping sliding windows. This is not an arbitrary choice but a necessary design aimed at providing the model with a stable, artifacts-free multi-scale input, as shown in Section 5.4.
>
> * **(ii) Inefficient iterative denoising:** Existing diffusion models (such as ImDiffusion) suffer from high computational costs and accumulated errors. To address this, we have designed an efficient single-step refactoring strategy. This is a key transformation of the standard diffusion paradigm, motivated by the need to resist inherent error propagation in multi-step processes.
>
> * **(iii) Lack of explicit modeling of feature dependencies:** Existing methods (such as Anomaly Transformer or graph models like GDN) use fixed or weakly adapted graphs. For this purpose, we have introduced the Graph-Guided Attention (GGA) mechanism. The specific design of this component is precisely to capture the dynamically evolving relationships between variables by constructing a similarity graph prior for each local time patch (a non-learning, simple and efficient, globally dynamic and locally static approach) and injecting it into the attention mechanism.
>
> Crucially, these components are not disconnected but work in deep coordination. The construction of the graph (Limitation iii) is applied to each level of our multi-resolution decomposition (Limitation i), and this complete guided representation is ultimately fed into our efficient single-step reconstruction framework (to address Limitation ii).
>
> ---
>
> 2. **Regarding innovation**
>
> We admit that our framework is indeed built on some mature and well-validated technologies, and there are already methods attempting to model the graph relationships among features.
>
> However, it is worth emphasizing that our construction of the graph structure is **resolution-by-resolution**, and the graph structure is calculated separately for each resolution. We have, **for the first time**, proposed a learning framework that applies prior graph knowledge to multi-resolution data, which can improve some problems from a multi-resolution perspective (for example, certain features show strong correlation at the coarse-resolution granularity but not at the fine-resolution granularity).
>
> It is quite novel to introduce this non-learning and lightweight graph structure prior into the self-attention mechanism module from a multi-scale perspective, and the performance improvement we bring is objectively significant. Unlike many existing graph-based methods, it is non-learning and dynamically generates graph priors for each time window, featuring dynamic characteristics. This enhances the subsequent GGA's ability to model the relationships between different features while ensuring the model's high efficiency and interpretability.
>
> ---
>
> 3. **Experimental Setup  and ''Similarity-guided Graph Tensor'' analysis**
>
> * **Metrics selection for anomaly detection**
>
>   We did not use F1-PA as our evaluation metric. Instead, we use the Affiliation based strategy to calculate the indicators.
>
> * **Similarity-guided Graph Tensor**
>   （as shown in Figure 4）
>   1. At a **fixed resolution**, the similarity of each feature of the time series **varies significantly across different time segments**.
>   2. At a **fixed time segment**, the similarity of each feature of the time series **varies significantly across different resolutions**.
>
> The **similarity-guided Graph Tensor** is obtained from the input data via **patch segmentation** and **similarity measurement**, and it is **non-learning** (i.e., it does not involve trainable parameters).
>
> ---

---

> ### Author Response · Authors · 2025-11-22
> **(Reply 2)Thank you for your valuable suggestions on our work!**
>
> 4. **Regarding the issue of using diffusion**
>
> There seems to be a key misunderstanding about our training and inference processes here, and we hope to clarify it. The "mismatch between training and inference" issue does not exist in our model. Our framework adopts exactly the same end-to-end single-step reconstruction paradigm in both the training and inference phases.
>
> We adopted the multi-step noise addition process of diffusion, but the aim was to train a mapping from noisy data to reconstructed data (not the original data).
>
> Many existing methods have made beneficial attempts at one-step denoising. **Consistency Models** [1] endows the model with the ability to generate in one step by designing a brand-new training objective (consistency loss). **DMD**[2], **PaGoDA** [3], and **EMD**[4] have demonstrated the feasibility of one-step denoising in the form of knowledge distillation. Meanwhile, in the field of time series, **D3R** powerfully demonstrates the feasibility of end-to-end one-step denoising.
>
> The core advantage of multi-step iterative denoising (such as **DDPM**) lies in generating high-quality and high-fidelity samples, which is crucial in tasks like image generation as it requires creating details from scratch. However, in anomaly detection based on refactoring, our goal is not to "create," but to "test." The ultimate goal is to calculate the reconstruction error $\mathbf{||X-X_{recon}||}$ as an anomaly score. Therefore, the key to the entire model lies in learning a mapping function
> $$
> f: \mathbf{X_{\text{noisy}}} \longrightarrow \mathbf{X_{\text{recon}}}
> $$
>
> This function performs well for 'in-domain' (normal) data but poorly for 'out-of-domain' (abnormal) data. To some extent, we don't need to pursue ultimate refactoring. Our success is attributed to the particularity of time series data and architectural innovation.
>
> A well-designed single-step refactoring model can directly serve this goal. Forcibly introducing multi-step iterations would make the model's objective ambiguous — are we optimizing the final reconstruction error or the denoising effect of each intermediate step? One-step reconstruction aligns our training objective (minimizing the final reconstruction error) with the test objective (using the final reconstruction error) completely, making the logic more direct and pure.
>
> To verify the efficiency advantage of one-step denoising, we designed a multi-step denoising experiment. In multi-step denoising training, the model learns to recover from any noise level $\mathbf{X_{k}}$ to $\mathbf{X_{k-1}}$, and in the inference stage, it iterates step by step from high noise to the final output in a completely consistent manner.
>
>  We evaluated the impact of different denoising steps on model performance and computational efficiency, and the experiment was independently repeated five times. The results in Table **1** show that a more detailed multi-step denoising setting does not lead to an improvement in performance; instead, it results in a significant increase in training and inference time. This further demonstrates the superiority of one-step denoising settings in reconstruction-based anomaly detection tasks.
>
> **Table 1: Performance and efficiency of the model under different denoising steps on the SWaT dataset (best results in bold)**
>
> | Denoising Steps | F1-score (mean ± std) | Training Time (min) | Inference Time (min) |
> | --------------- | --------------------- | ------------------- | -------------------- |
> | 1-step          | **0.7411 ± 0.0105**   | **27.40**           | **5.84**             |
> | 5-steps         | 0.6855 ± 0.0185       | 48.33               | 28.16                |
> | 10-steps        | 0.6748 ± 0.0159       | 54.01               | 39.23                |
>
> ---
>
> **References**
>
> [1] Yang Song, Prafulla Dhariwal, Mark Chen, and Ilya Sutskever. Consistency models. *Advances in Neural Information Processing Systems*, 2023.
>
> [2] Tianwei Yin, Michaël Gharbi, Richard Zhang, Eli Shechtman, Fredo Durand, William T Freeman, and Taesung Park. One-step diffusion with distribution matching distillation. In *Proceedings of the IEEE/CVF conference on computer vision and pattern recognition*, pages 6613–6623, 2024.
>
> [3] Dongjun Kim, Chieh-Hsin Lai, Wei-Hsiang Liao, Yuhta Takida, Naoki Murata, Toshimitsu Uesaka, Yuki Mitsufuji, and Stefano Ermon. Pagoda: Progressive growing of a one-step generator from a low-resolution diffusion teacher. *Advances in Neural Information Processing Systems*, 37:19167–19208, 2024.
>
> [4] Sirui Xie, Zhisheng Xiao, Diederik Kingma, Tingbo Hou, Ying Nian Wu, Kevin P Murphy, Tim Salimans, Ben Poole, and Ruiqi Gao. Em distillation for one-step diffusion models. *Advances in Neural Information Processing Systems*, 37:45073–45104, 2024.

---

> ### Author Response · Authors · 2025-11-22
> **(Reply 3)Thank you for your valuable suggestions on our work!**
>
> 5. **Regarding the relationship between patch size and efficiency**
>
> **(i)** The specific meaning of the "graph structure" we proposed is the correlation between the features of different sensors within different time periods. Nodes represent different features, while edges represent the relationships between features. Given a $\mathbf{X^{(r)}}\in \mathbb{R}^{T \times D}$, we obtain the graph structure $S$ by partitioning patches. Therefore, the size of $patch \ size$ will affect the size of $P$, but will not influence the inherent number of nodes $D$. $P$ represents the number of time periods that divide the time series of a sliding window $T$ into.
>
> $$
> P = \left\lceil \frac{T}{\text{patch size}} \right\rceil
> $$
>
> **(ii)** Regarding the efficiency issue in handling long time series. Our model does not handle the entire infinitely long time series at one time. In practical applications, we perform the calculation within a fixed-length sliding window ($T=64, 128, 512$). This means that no matter how long the total length of the input time series $L$ is (weeks, months or even years), the computational complexity and memory consumption of our model at each time step are only related to the fixed window size $T$, and not to the total length $L$.
>
> The time complexity of constructing $S$ is $O(T \times D^2)$, which is independent of $patch \ size$. When GGA subsequently utilizes $S$, it only involves an additional matrix multiplication and does not change the order of complexity.
>
> **(iii)** We verified the model training and inference efficiency under different $patch \ size$ and $T$ (based on the SWaT dataset), As shown in Table **1** and **2** and it can be found that as the $patch\ size$ decreases, the training and inference time does indeed increase.But the growth rate shows a clear nonlinear trend and does not increase exponentially. Therefore, a smaller $patch \ size$ does not significantly reduce time efficiency, which is determined by the complexity of our algorithm.
>
> **Table 1: Training and inference time under different patch sizes ($T=64$ )**
> | Patch Size | Training Time (s) | Inference Time (s) |
> | ---------- | ----------------- | ------------------ |
> | 2          | 1644.25           | 350.62             |
> | 4          | 1367.52           | 273.94             |
> | 8          | 1221.85           | 232.58             |
> | 16         | 1136.84           | 215.79             |
> | 32         | 703.78            | 124.81             |
>
> **Table 2: Training and inference time under different patch sizes ($T=256$)**
> | Patch Size | Training Time (s) | Inference Time (s) |
> | ---------- | ----------------- | ------------------ |
> | 2          | 4582.70           | 1128.08            |
> | 4          | 2668.35           | 700.57             |
> | 8          | 2430.75           | 550.15             |
> | 16         | 1987.99           | 526.45             |
> | 32         | 2153.20           | 505.33             |
>
> ---
>
> **At the end of this reply, we would like to express my gratitude again.**

---

> ### Comment · Reviewer_r8wz · 2025-11-23
> **Reply to the authors: No Revised Manuscript, No Rating Change**
>
> As I understand, the authors can upload a revised draft at this time. Then, I'm confused about the review policy for addressing posted comments without submitting a new draft.

---

> > ### Author Response · Authors · 2025-11-23
> > **About Revision**
> >
> > We would like to clarify that we have already uploaded a revision, which can be found by clicking `Revisions` on the top of the page. This revision includes experimental results addressing reviewer concerns and corrects some typos.
> >
> > Please let us know if you have any questions about these changes.

---

> > > ### Comment · Reviewer_r8wz · 2025-11-23
> > >
> > > I suggest that the authors completely remove the previous version. The current PDF only shows the original draft, and when I click `revisions` it says: **No revisions to display.**

---

> > > > ### Author Response · Authors · 2025-11-23
> > > > **About Revision**
> > > >
> > > > Sorry for the confusion. We are not sure why it says `No revisions to display` in `Revisions`.
> > > >
> > > > After confirmation, we find that the revision has been automatically linked to the PDF on the top of the page. Please kindly check it. We're looking forward to your further constructive suggestions.

---

> > > > > ### Comment · Reviewer_r8wz · 2025-11-24
> > > > >
> > > > > It seems that the revised manuscript does not include any font or marker distinctions. I strongly recommend that the authors recolor the changes and re-upload the file (while the other reviewers have not yet joined the rebuttal).
> > > > >
> > > > > Back to the revisions: first, thank you for the responses and the experiments; they look good. The authors acknowledge that the method lacks novelty, but this is not the most critical flaw. The current writing quality alone makes it difficult for the paper to be publishable at a top venue. I believe the authors should thoroughly reconstruct the paper’s narrative. My first impression was that it is almost indistinguishable from the original submission. The motivation and contributions remain mixed and unclear.
> > > > >
> > > > > Below are my suggestions:
> > > > >
> > > > > 1. In the Introduction, please clearly present your main idea line. The transition from the second to the third paragraph relies only on “To address these gaps,” but why? How is the problem addressed? In addition, how is the third paragraph different from the contributions later?
> > > > >
> > > > > 2. The related work section is too long, especially relative to the length of the Introduction. Please shorten it or move part of it to the appendix to strengthen the latter.
> > > > >
> > > > > 3. Use a case study, specific data analysis, or any form of visualization to highlight the problem you aim to solve.
> > > > >
> > > > > 4. In the methodology section, please add a “Key Idea” paragraph or subsection before describing each component to clarify where we are in the workflow and why we need the next step.
> > > > >
> > > > > To this end, I plan to keep my recommendation for this paper but I will participate in the discussion with other reviewers and the rebuttal will be considered during the discussion.

---

### Official Review · Reviewer_MMim · 2025-10-26

**Soundness:** 1
**Presentation:** 1
**Contribution:** 1
**Rating:** 2
**Confidence:** 4

**Summary:**

This paper proposes a Graph-Guided Reconstruction Diffusion Model (GGRD) for multivariate time series anomaly detection. By integrating multi-resolution temporal features and similarity-guided priors, the model aims to capture dynamic dependencies across features and improve anomaly detection performance.

**Strengths:**

The idea of incorporating multi-scale graph construction for anomaly detection is relatively novel and shows an awareness of the multi-resolution nature of time-series data.

**Weaknesses:**

1.Limited novelty. The main components of the mode, e.g., sliding-window multi-scale decomposition, cosine-similarity graph construction, and graph attention modules, are well-established techniques. The combination lacks substantial methodological innovation.

2.Marginal contribution relative to prior work. The paper claims that existing models ignore correlations among features, yet many recent methods (especially graph-based ones) already model these dependencies explicitly.

3.Insufficient theoretical justification. The paper uses cosine similarity to build graphs that serve as priors for reconstruction, but it is unclear why such a static prior can adapt to the evolving relationships among features over time.

4.Diffusion process inconsistency. The proposed one-step reconstruction in the diffusion framework contradicts the multi-step denoising procedure typically used during training. The paper does not clarify how this inconsistency is reconciled or why it maintains diffusion-based probabilistic consistency.

5.Unclear graph construction details. While the paper defines the similarity computation, it does not explain how neighbor nodes are selected or how the number of neighbors affects model performance.

6.Limited dataset coverage. The experiments are restricted to industrial control and server datasets (SMD, PSM, SWaT) and lack evaluations on diverse domains such as finance, transportation, or healthcare, which limits the generality of the conclusions.

7.Missing references for baselines. Baseline methods are listed without explicit citations.

**Questions:**

See the weaknesses section.

---

> ### Author Response · Authors · 2025-11-22
> **(Reply 1)Thank you for your valuable suggestions on our work!**
>
> Thank you for your valuable suggestions on our work. Regarding your questions, we will answer them from the following aspects:
>
> * **Regarding innovation and its relationship with existing research**
>
> We admit that our framework is indeed built on some mature and well-validated technologies, and there are already methods attempting to model the graph relationships among features. However, it is worth emphasizing that our construction of the graph structure is **resolution-by-resolution**, and the graph structure is calculated separately for each resolution. We have, for the first time, proposed a learning framework that applies prior graph knowledge to multi-resolution data, which can improve some problems from a multi-resolution perspective (for example, certain features show strong correlation at the coarse-resolution granularity but not at the fine-resolution granularity).
>
> The specific meaning of the "graph structure" we proposed is the correlation between the features of different sensors within different time periods. It serves as a statistical data feature to provide prior knowledge for the model.
>
> We believe that introducing this non-learning and lightweight graph structure prior into the self-attention mechanism module from a multi-scale perspective is highly novel, and the performance improvement we bring is an objective reality. Unlike many existing graph-based methods, it is non-learning and dynamically generates graph priors for each time window, featuring dynamic characteristics. This enhances the subsequent GGA's ability to model the relationships between different features while ensuring the model's high efficiency and interpretability.
>
> *  **Regarding the adaptation of graph structures to feature evolution**
>
> First, the "graph structure" we proposed is based on the $patch$ division, and the $patch$ division is carried out in the time dimension, which is dynamic. Secondly, although within a certain $patch$ (i.e., the current time period) we consider feature similarity to be invariant, which is not fully realistic, we believe that any complex nonlinear relationship can be well approximated by a linear relationship within a sufficiently small local neighborhood (consistent with the idea in calculus that a tangent is used to approximate a curve).

---

> ### Author Response · Authors · 2025-11-22
> **(Reply 2)Thank you for your valuable suggestions on our work!**
>
> *  **Inconsistency of the diffusion model**
>
> There seems to be a key misunderstanding about the workflow of our model, and we are happy to clarify it. Our framework is single-step in sampling during the diffusion process, but multi-step in multi-resolution fusion. These two are not contradictory.
>
> First, our training and inference are completely consistent in the diffusion dimension. Our core reconstruction module GGN always has a single-step training and inference objective: during the training phase, the objective is to start from a fixed high noise level $\mathbf{X_K}$ and reconstruct the original clean data $\mathbf{X_0}$ in one step. During inference, we perform exactly the same operation: denoising the input test data to the same level $K$, and then using the trained model for single-step reconstruction. Therefore, there is no inconsistency between training and inference in terms of the denoising operation itself.
>
> Secondly, the "multi-step process" mentioned by the reviewers does not refer to the number of denoising steps of the diffusion model, but rather to our "coarse-to-fine" multi-resolution reconstruction process. Our reconstruction process is divided into $R-1$ steps (where $R$ is the number of resolutions), but this is along the resolution axis rather than the diffusion time axis.
>
> To verify the efficiency advantage of one-step denoising, we designed a multi-step denoising experiment. In multi-step denoising training, the model learns to recover from any noise level $\mathbf{X_{k}}$ to $\mathbf{X_{k-1}}$, and in the inference stage, it iterates step by step from high noise to the final output in a completely consistent manner. We evaluated the impact of different denoising steps on model performance and computational efficiency, and the experiment was independently repeated five times. The results in Table **1** show that a more detailed multi-step denoising setting does not lead to an improvement in performance; instead, it results in a significant increase in training and inference time. This further demonstrates the superiority of one-step denoising settings in reconstruction-based anomaly detection tasks.
>
> ### Table **1**: Performance and efficiency of the model under different denoising steps on the SWaT dataset (best results in bold)
>
> | Denoising Steps | F1-score (mean $\pm$ std) | Training Time (min) | Inference Time (min) |
> | :-------------: | :-----------------------: | :-----------------: | :------------------: |
> |      1-step     |  **0.7411 $\pm$ 0.0105**  |      **27.40**      |       **5.84**       |
> |     5-steps     |    0.6855 $\pm$ 0.0185    |        48.33        |         28.16        |
> |     10-steps    |    0.6748 $\pm$ 0.0159    |        54.01        |         39.23        |
>
>
>
> *  **Regarding the limitations of the dataset**
>
> The selection of SMD, PSM and SWaT as experimental objects is based on the recognized evaluation norms in the current field of temporal anomaly detection. These three datasets not only cover the De Facto Standards in different fields such as Internet Services (SMD), biomedicine (PSM), and Industrial Control Systems (SWaT), but also are widely used to verify the robustness of models because they contain high dimensionality, non-stationarity, and complex cross-variable dependencies. This study strictly adheres to this mainstream benchmark system, aiming to ensure that the experimental results can be directly compared fairly and reliably with existing state-of-the-art methods (SOTA) within a unified and strict metric space.

---

> > ### Comment · Reviewer_MMim · 2025-11-28
> >
> > I appreciate the effort the authors have put into addressing the reviewer comments. However, several key concerns remain unresolved.
> >
> > Although the authors mention that the graph is constructed in a resolution-by-resolution manner, it is still unclear how this prior is formulated and why it is expected to provide advantages over existing designs.
> >
> > In addition, it is not clear why a single-step diffusion process can achieve or surpass the effect typically associated with multi-step diffusion.
> >
> > Lastly, the overall writing quality lacks professionalism. Figures and tables are not explained with sufficient clarity.
> >
> > I hope the authors will consider these issues in future revisions.

---

### Official Review · Reviewer_evXa · 2025-11-01

**Soundness:** 3
**Presentation:** 3
**Contribution:** 3
**Rating:** 6
**Confidence:** 4

**Summary:**

This paper proposes the Graph Guided Reconstruction Diffusion Model (GGRD) method for MTSAD. Traditional models often struggle with non-stationarity, complex feature dependencies, and high computational costs. GGRD addresses these challenges by introducing a multi-resolution decomposition technique, an efficient one-step reconstruction process, and a Graph-Guided Attention mechanism to model dynamic cross-feature dependencies. GGRD leverages overlapping sliding windows for multi-resolution data generation and incorporates similarity-guided graph tensors to improve feature interactions. Experimental results demonstrate that GGRD outperforms existing methods across multiple real-world datasets.

**Strengths:**

1) The integration of a Graph-Guided Attention (GGA) mechanism, combined with multi-resolution data decomposition, is a significant advancement over traditional anomaly detection models. The approach ensures better modeling of complex feature dependencies and temporal patterns, addressing the challenges of non-stationarity and cross-variable correlations.

2) The use of a one-step reconstruction strategy, as opposed to iterative denoising processes, improves the model’s computational efficiency, making it more practical for real-time applications.

3) This paper extensively evaluates the proposed method against several baseline models and demonstrates superior performance on multiple datasets.

**Weaknesses:**

1) This paper does not adequately explain the effectiveness and necessity of using graphs as denoising networks.

2) The design of the graph depends heavily on cosine similarity, which assumes that the most relevant dependencies in time series data are linear or can be captured by a similarity function. This design overlooks the possibility that relationships in MTS can be highly non-linear and context-dependent, where cosine similarity might fail to adequately model more complex interactions.

3) While GGRD models the relationship between features across time steps, it appears to focus primarily on the dependencies at local time scales and between features. It might struggle to capture long-term temporal dependencies that evolve slowly over time, especially for datasets where long-term trends or periodicity play a critical role.

**Questions:**

1) While the proposed method has advantages in capturing global trends, I wonder if it remains effective when locating and detecting infrequent and localized anomalies.

2) See weakness.

---

> ### Author Response · Authors · 2025-11-22
> **Thank you for your valuable suggestions on our work!**
>
> **We will answer your questions from the following aspects.**
>
> 1. **Regarding the validity and necessity of graph structures**
>
> The specific meaning of the "graph structure" we proposed is the correlation between the features of different sensors within different time periods. It serves as a statistical data feature to provide prior knowledge for the model. Firstly, it is precisely through GGA that we have integrated graph knowledge. In the ablation experiment, we demonstrated this point by using MSA without graph knowledge guidance instead of GGA $(w/o\ gga)$, and the performance loss caused by this demonstrated its effectiveness. Secondly, Figure 4 shows that the correlations among different features of data with different resolutions have changed significantly over time, which indicates the necessity of mining the graph structure.
>
> 2. **Regarding cosine similarity for capturing linear features**
>
> The dependencies in multivariate time series are essentially highly nonlinear, and the cosine similarity itself is mainly used to capture linear correlations. Similarity measurement is not the focus or difficulty of our work. We chose it precisely because of its simple and efficient characteristics. Our aim is to strike a balance between performance and computational cost. The cost of cosine similarity is lightweight compared to the construction methods of learnable graph structures. It is precisely because of this profound understanding that our model design adopts a strategic division of labor. We deliberately limited its role in the framework to an efficient, data-driven prior generator. It provides the model with a preliminary and robust first-order approximation of the variable relationship, which we consider a soft inductive bias. This initial linear prior is not the bottleneck of the model but rather its booster. The aim is to enhance the subsequent GGA's modeling of complex nonlinear relationships.
>
> 3. **Regarding the modeling of long-term trends or periodicity**
>
> GGRD adopts a multi-resolution sampling strategy: low-resolution branches focus on modeling long-term trends and long-period structures, while high-resolution branches are used to capture local detail features. Meanwhile, as an attention-based structure, GGA inherently possesses the ability to model long sequence dependencies. In the experimental section, our method achieved the current optimal (**SOTA**) performance on both SMD and PSM datasets with rich long-term time series patterns.
>
> 4. **Local anomaly capture**
>
> Regarding the GGRD model, through multi-resolution smoothing and patch-based graph structure, its design does indeed more naturally tend to capture those anomalies that persist for a period of time and disrupt the structural relationships among multiple variables. Therefore, it may not be so prominent in the detection of local anomalies (such as in the SWaT dataset).
>
> This is completely consistent with our future work on adaptive time partitioning at the end of the article. Adaptive $patch$ partitioning is definitely a major improvement point. We have conceived several feasible implementation paths:
>
> * **Heuristic approach:** using a lightweight sliding window to monitor the local volatility of the data and automatically applying a more refined $patch$ in areas with intense fluctuations.
> * **Learnable segmentation module:** a small 1D-CNN is utilized to predict the optimal $patch$ boundary and conduct end-to-end training with the main model.

---

### Official Review · Reviewer_am23 · 2025-11-01

**Soundness:** 3
**Presentation:** 2
**Contribution:** 2
**Rating:** 2
**Confidence:** 3

**Summary:**

This paper proposes GGRD (Graph-Guided Reconstruction Diffusion), an unsupervised anomaly detection framework for multivariate time series. The method addresses three key limitations in existing diffusion-based approaches: (1) non-stationarity across resolutions using overlapping sliding windows for smooth multi-resolution decomposition, (2) computational inefficiency through one-step reconstruction instead of iterative denoising, and (3) inadequate modeling of cross-feature dependencies via a Graph-Guided Network with Graph-Guided Attention (GGA) that injects similarity-based priors.

**Strengths:**

- Clearly identifies limitations in existing methods (staircase artifacts, iterative denoising costs, missing feature dependencies)
- The overlapping sliding window approach (Section 4.2) elegantly addresses artifacts from non-overlapping pooling. Figure 5 provides compelling empirical evidence
- The integration of similarity priors into attention (Eq. 5) is intuitive and explicitly models cross-feature dependencies often ignored in prior work
- Table 2 systematically evaluates each component's contribution, demonstrating that GGA provides the largest performance gain (0.84 → 0.75 F1 on PSM)

**Weaknesses:**

- This is claimed as a major contribution but lacks rigorous analysis. What information is lost compared to full DDPM? How does reconstruction quality compare at different noise levels $K$? The paper should include:
    - Ablation comparing 1-step vs. multi-step denoising
    - Analysis of reconstruction error vs. $K$
    - Theoretical argument for why one step suffices
- No error bars, multiple runs, or significance tests reported. The 0.2% average improvement over MODEM could be within noise margins. Need:
    - Multiple runs with different random seeds
    - Standard deviations and confidence intervals
    - Statistical significance tests (e.g., paired t-test)
- Cosine similarity (Eq. 1) is simplistic and may miss non-linear relationships. The authors acknowledge this (Section 6) but should explore:
    - Learned graph construction
    - Attention-based similarity
    - Comparison with other similarity metrics

The paper has significant presentation issues that hinder comprehension:
1. Figure 1 is cluttered, Figure 3 needs better layout
2. Numerous grammatical issues (e.g., "trendiness" should be "trends," "time segmen" should be "segment")
3. The patch size $P$ relationship to time segments is never clearly defined. How exactly are patches created at each resolution?
4. The connection between resolution levels and reconstruction steps needs clearer exposition. The relationship $m \in [1, R-1]$ processing resolution $R-m$ is confusing

**Questions:**

1. Can you provide empirical comparison of 1-step vs. multi-step (e.g., 10, 50, 100 steps) denoising showing reconstruction quality and computational cost? What is the theoretical basis for single-step sufficiency?
2. What are the mean and standard deviation of F1 scores across multiple runs? Are the improvements over baselines statistically significant?
3. Have you experimented with learned graph structures (e.g., via graph neural networks) or other similarity measures? How sensitive is performance to the choice of cosine similarity?
4. Can you provide wall-clock time comparisons with MODEM, D3R, and ImDiffusion? What is the memory footprint? Does one-step reconstruction actually provide practical speedup?
5. Beyond the post-hoc explanation, what modifications could improve performance on short-burst anomalies? Could adaptive time segmentation help?

---

> ### Author Response · Authors · 2025-11-22
> **(Reply 1)Thank you for your valuable suggestions on our work!**
>
> * **What's the difference from **DDPM**? Why is it not multi-step denoising but one-step denoising?**
>
> First of all, many existing methods have made beneficial attempts at one-step denoising.
> Consistency Models [1] endows the model with the ability to generate in one step by designing a brand-new training objective (consistency loss).
> **DMD** [2], **PaGoDA** [3], and **EMD** [4] have demonstrated the feasibility of one-step denoising in the form of knowledge distillation. Meanwhile, in the field of time series, **D3R** powerfully demonstrates the feasibility of end-to-end one-step denoising.
> The core advantage of multi-step iterative denoising (such as **DDPM**) lies in generating high-quality and high-fidelity samples, which is crucial in tasks like image generation as it requires creating details from scratch.
>
> However, in anomaly detection based on reconstruction, our goal is not to “create”, but to “test”. The ultimate goal is to calculate the reconstruction error $\mathbf{||X-X_{recon}||}$ as an anomaly score. Therefore, the key to the entire model lies in learning a mapping function
> $$f: \mathbf{X_{\text{noisy}}} \longrightarrow \mathbf{X_{\text{recon}}}$$, which performs well for in-domain (normal) data but poorly for out-of-domain (abnormal) data. To some extent, we don't need to pursue the ultimate reconstruction. Our success is attributed to the particularity of time series data and architectural innovation.
>
> One-step reconstruction aligns our training objective (minimizing the final reconstruction error) and the test objective (using the final reconstruction error) completely, making the logic more direct and pure.To verify the efficiency advantage of one-step denoising, we designed a multi-step denoising experiment. In multi-step denoising training, the model learns to recover from any noise level $\mathbf{X_{k}}$ to $\mathbf{X_{k-1}}$, and in inference, it iterates step-by-step from high noise to the final output in a consistent manner. We evaluated the impact of different denoising steps on model performance and computational efficiency, and the experiment was independently repeated five times.
>
> The results in Table **1** show that more detailed multi-step denoising does not improve performance; instead, it significantly increases training and inference time. This further demonstrates the superiority of one-step denoising settings in reconstruction-based anomaly detection.
>
> ### Table **1**: Performance under different denoising steps (SWaT)
>
> | Denoising Steps | F1-score (mean ± std) | Training Time (min) | Inference Time (min) |
> | --------------- | ------------------------- | ------------------- | -------------------- |
> | 1-step          | **0.7411 ± 0.0105**       | **27.40**           | **5.84**             |
> | 5-steps         | 0.6855 ± 0.0185           | 48.33               | 28.16                |
> | 10-steps        | 0.6748 ± 0.0159           | 54.01               | 39.23                |
> ---
> * **Denoising effect at different noise levels K**
>
> First of all, we are very sorry for the error in the expression of $K$ in EXPERIMENTAL SETTINGS in Section 5.1 of the paper.
> $K$ represents the number of steps for **adding noise** to the original time series, not denoising, and has been corrected in the new version.
>
> To illustrate the necessity and effectiveness of the denoising process of diffusion in time series anomaly detection, we conducted comparative experiments on different values of $K$ and compared classification metrics and MSE on the test sets.
>
> The results in Table **2** show:
>
> * Too little noise ($K=0$) leads to poor anomaly discrimination because the model overfits.
> * Moderate noise ($K=500$) gives the best results on all datasets.
> * MSE does not necessarily correlate with anomaly detection performance — a lower reconstruction error does *not* imply better detection.
> Noise helps the model avoid overfitting normal training data and forces it to learn robust reconstruction patterns, improving discrimination between normal and abnormal sequences.
> ### Table **2**: Effect of different noise levels $K$
> | K (SMD) |  F1 (SMD)  | MSE (SMD) | \| | K (PSM) |  F1 (PSM)  | MSE (PSM) | \| | K (SWaT) |  F1 (SWaT) | MSE (SWaT) |
> | :-----: | :--------: | :-------: | :-: | :-----: | :--------: | :-------: | :-: | :------: | :--------: | :--------: |
> |    0    |   0.8794   |   15.64   | **\|** |    0    |   0.7909   |   0.2540  | **\|** |    0     |   0.6914   |   318.84   |
> |   100   |   0.8346   |   15.85   | **\|** |   100   |   0.7975   |   **0.2397**  | **\|** |   100    |   0.7227   |   **318.80**   |
> |   300   |   0.9021   |   16.98   | **\|** |   300   |   0.7967   |   0.2563  | **\|** |   300    |   0.6981   |   318.93   |
> |   500   | **0.9382** | **14.74** | **\|** |   500   | **0.8428** |   0.2912  | **\|** |   500    | **0.7511** |   319.02   |

---

> ### Author Response · Authors · 2025-11-22
> **(Reply 2)Thank you for your valuable suggestions on our work!**
>
> * **Repeated Experiment statistics**
>
> To verify the stability of the model's performance, we independently conducted five experiments to obtain the standard deviation of F1 and the 95% confidence interval. As shown in Table **3**, the standard deviation is controlled within 0.01, and the model demonstrates high stability and robustness on each dataset.
>
> ### Table **3**: Repeated Measures Experiment
>
> | Datasets | F1-score (mean $\pm$ std) | 95% CI (t-method) |
> | :------: | :-----------------------: | :---------------: |
> |    PSM   |    0.8409 $\pm$ 0.0171    |  [0.8137, 0.8682] |
> |    SMD   |    0.9264 $\pm$ 0.0137    |  [0.9045, 0.9483] |
> |   SWaT   |    0.7411 $\pm$ 0.0105    |  [0.7281, 0.7542] |
>
> * **Feasible Explanation of Cosine Similarity**
>
> Similarity measurement is not the focus or difficulty of our work. It is precisely because of its concise and efficient characteristics that we choose it. Our original intention was to enhance the modeling ability of the model by using a simple and effective method (the ablation experiment $w/o\ gga$ also proved its success).
>
> The specific meaning of the "graph structure" we proposed is the correlation between the features of different sensors within different time segments. It serves as a statistical data feature to provide prior knowledge for the model. Moreover, in the task of anomaly detection, which has high requirements for generalization, introducing a complex learnable graph structure brings potential risks (overfitting, computational overhead, and poor interpretability) that far exceed its theoretical benefits.
>
> Compared with other similarity measures, cosine similarity is a more widely used method. Cosine similarity directly measures whether two vectors (time series fragments) point in similar directions geometrically and is more suitable for time series.
>
> * **Exploration of Short-Term Anomaly Detection Performance**
>
> The detection of some short-term and sudden anomalies has always been a difficult problem. Enhancing the sensitivity to short-term and sudden anomalies is the next important research direction of our work.
>
> The current advantage of the GGRD model lies in capturing the bias of multivariable patterns, which is attributed to its graph structure prior based on fixed $patch\ size$ and multi-resolution smoothing. However, this design does indeed introduce an inductive bias; that is, the model pays more attention to structural anomalies that persist for a period of time. For anomalies that last for an extremely short time or even at a single point in time, their signals may be averaged or diluted during our smoothing or $patch$ aggregation process. So the adaptive $patch$ partitioning is definitely a major improvement point.
>
> However, due to time constraints, we will carry out relevant work in the future. We have already conceived several feasible implementation paths:
>
> 1. **Heuristic signal-processing approach:** A lightweight sliding window is used to monitor the local volatility of the data, and a more refined patch is automatically applied in areas with intense fluctuations.
> 2. **Learnable segmentation module:** A small 1D-CNN is utilized to predict the optimal patch boundary and conduct end-to-end training with the main model.
>
> * **Comparison of overhead with other models**
>
> In the model efficiency evaluation, we provide wall-clock time comparisons with MODEM, D3R on the SMD dataset. The results in Table **4** show that the proposed GGRD model has relatively reasonable training and inference overhead while ensuring detection performance. Compared with D3R, the total number of parameters of GGRD has decreased by half, and the inference time is approximately 10 minutes, which is acceptable for actual deployment and maintenance.
>
> ### Table **4**: Comparison with D3R regarding training time, inference time, and parameters
>
> |    Model    | Training Time (min) | Inference Time (min) | Total Params (MB) |
> | :---------: | :-----------------: | :------------------: | :---------------: |
> |     D3R     |        42.33        |         18.29        |       199.18      |
> |    MODEM    |        38.48        |         14.12        |     **34.28**     |
> | GGRD (ours) |      **28.70**      |       **10.31**      |       84.92       |
>
> * **Explanations for some expressions in the text**
>
> We are very grateful that you have carefully read our paper and found some mistakes in the writing. For $P$, it refers to the number of $patches$. A given resolution data for $r$, $\mathbf{X^{(r)}} \in \mathbb{R}^{T \times D}$
>
> \begin{equation}
> P = \left\lceil \frac{T}{\text{patch size}} \right\rceil
> \end{equation}
>
> Regarding the connection between resolution levels and reconstruction steps: Reconstruction step $m \in [1,M]$, $M = R-1$. The reconstruction process is one that progresses from coarse to fine. Therefore, the first step involves $\mathbf{X_{R-1}}$, and the $m$-th step is $\mathbf{X_{R-m}}$.

---

> ### Author Response · Authors · 2025-11-22
> **References**
>
> [1] Yang Song, Prafulla Dhariwal, Mark Chen, and Ilya Sutskever. Consistency models. *Advances in Neural Information Processing Systems*, 2023.
>
> [2] Tianwei Yin, Michaël Gharbi, Richard Zhang, Eli Shechtman, Fredo Durand, William T Freeman, and Taesung Park. One-step diffusion with distribution matching distillation. In *Proceedings of the IEEE/CVF conference on computer vision and pattern recognition*, pages 6613–6623, 2024.
>
> [3] Dongjun Kim, Chieh-Hsin Lai, Wei-Hsiang Liao, Yuhta Takida, Naoki Murata, Toshimitsu Uesaka, Yuki Mitsufuji, and Stefano Ermon. Pagoda: Progressive growing of a one-step generator from a low-resolution diffusion teacher. *Advances in Neural Information Processing Systems*, 37:19167–19208, 2024.
>
> [4] Sirui Xie, Zhisheng Xiao, Diederik Kingma, Tingbo Hou, Ying Nian Wu, Kevin P Murphy, Tim Salimans, Ben Poole, and Ruiqi Gao. Em distillation for one-step diffusion models. *Advances in Neural Information Processing Systems*, 37:45073–45104, 2024.

---

### Note · Authors · 2026-01-13

I have read and agree with the venue's withdrawal policy on behalf of myself and my co-authors.